# Using Telehealth to Guarantee the Continuity of Rehabilitation during the COVID-19 Pandemic: A Systematic Review

**DOI:** 10.3390/ijerph191610325

**Published:** 2022-08-19

**Authors:** Elisabetta Brigo, Aki Rintala, Oyéné Kossi, Fabian Verwaest, Olivier Vanhoof, Peter Feys, Bruno Bonnechère

**Affiliations:** 1REVAL Rehabilitation Research Center, Faculty of Rehabilitation Sciences, Hasselt University, 3590 Diepenbeek, Belgium; 2Faculty of Social Services and Health Care, LAB University of Applied Sciences, 15210 Lahti, Finland; 3ENATSE, National School of Public Health and Epidemiology, University of Parakou, Parakou 03 BP 10, Benin

**Keywords:** telerehabilitation, telehealth, rehabilitation, physiotherapy, technologies, COVID-19

## Abstract

COVID-19 has abruptly disrupted healthcare services; however, the continuity of rehabilitation could be guaranteed using mobile technologies. This review aims to analyze the feasibility and effectiveness of telehealth solutions proposed to guarantee the continuity of rehabilitation during the COVID-19 pandemic. The PubMed, Cochrane Library, Web of Science and PEDro databases were searched; the search was limited to randomized controlled trials, observational and explorative studies published up to 31 May 2022, assessing the feasibility and effectiveness of telerehabilitation during the COVID-19 pandemic. Twenty studies were included, for a total of 224,806 subjects: 93.1% with orthopedic complaints and 6.9% with non-orthopedic ones. The main strategies used were video and audio calls via commonly available technologies and free videoconferencing tools. Based on the current evidence, it is suggested that telerehabilitation is a feasible and effective solution, allowing the continuity of rehabilitation while reducing the risk of infection and the burden of travel. However, it is not widely used in clinical settings, and definitive conclusions cannot be currently drawn. Telerehabilitation seems a feasible and safe option to remotely deliver rehabilitation using commonly available mobile technologies, guaranteeing the continuity of care while respecting social distancing. Further research is, however, needed to strengthen and confirm these findings.

## 1. Introduction

During the coronavirus disease 2019 (COVID-19) pandemic, social distancing policies have been established in an attempt to prevent the spread of the virus, resulting in the disruption of many healthcare services; one of the most severely affected ones has been rehabilitation care [1]. The WHO defines rehabilitation as a series of interventions aimed to optimize the functioning of individuals in order to allow them to interact with their environment and engage in activities meaningful to them. Rehabilitation is not a luxury nor an optional health service; timely, affordable and high-quality rehabilitation care should be available to anyone who needs it [2]. According to the WHO estimations, there are currently about 2.4 billion people in the world that would require and benefit from rehabilitation, and unfortunately, such need is nowadays largely unmet [2]. As a consequence of the COVID-19 crisis, organizations had to rethink how to deliver healthcare services, and an alternative care model has thus emerged, leveraging information and communication technologies (ICT) to guarantee the continuance of such services. Health services delivered via digital means are referred to using terms such as “telehealth”, “eHealth” or “mHealth” [3,4]; with specific regard to physiotherapy, the term “telerehabilitation” has been widely used in the literature to describe rehabilitation services delivered via ICT [5,6], and this is the term we will also use from this point on in the present review. Telerehabilitation can be delivered via a variety of digital means, either with synchronous audio and/or video calls or with asynchronous media, such as recorded videos, text messages, emails and links to educational materials [6,7]. 

Telerehabilitation was already in use before the current pandemic, as also reported by a previous systematic review through a meta-analysis by Cottrell et al. (2017) [6], including studies from inception up to 2015. The results from this review suggested that telerehabilitation seems superior (or at least not inferior) to standard face-to-face physiotherapy practice for a variety of musculoskeletal disorders, thus making telerehabilitation a viable option for the clinical management of such conditions.

The COVID-19 pandemic has expedited the development and implementation of telehealth, with the number of healthcare interventions delivered via digital devices increasing exponentially, also thanks to the wide availability of mobile technology. In fact, the number of smartphone subscriptions worldwide today has reached six billion and has been estimated to further grow by several hundred million in the next few years [8]. This may open up new perspectives and opportunities in the healthcare sector, with previous studies already highlighting the patients’ acceptance of telehealth, increasing adherence [9,10] and patient satisfaction [11,12,13]. Telerehabilitation could indeed complement the current rehabilitation services, allowing healthcare professionals to support and treat patients in remote locations using telecommunications technology, guaranteeing people access to medical expertise in a quick, flexible and efficient manner, without having to travel, resulting in less burden and great satisfaction. However, the current level of evidence supporting this kind of intervention in clinical practice is still relatively limited [14]. Considering that the actual extent of the implementation of telerehabilitation during the current COVID-19 pandemic is still unclear, this review aims to summarize and analyze the different solutions that have been proposed to remotely support patients with a variety of health conditions in their rehabilitation process during the COVID-19 pandemic, highlighting the effectiveness and feasibility of such interventions.

## 2. Materials and Methods

The protocol of the present study was registered in the International Prospective Register Of Systematic Reviews PROSPERO (registration number CRD42021257073). 

### 2.1. Search Strategy

We searched the PubMed electronic database of the National Library of Medicine, the Web of Science database, the Cochrane Library and the PEDro database for relevant articles published up to 31 May 2022; MeSH terms and keywords referring to telerehabilitation (e.g., telerehabilitation, telephysiotherapy, remote rehabilitation, remote physiotherapy) and COVID-19 (e.g., COVID, COVID-19, SARS-CoV-2) were used for this search (see Appendix A for more details regarding the search strategies applied to each of the aforementioned databases). The references from the selected papers and relevant articles were screened for potential additional studies, in accordance with the snowball principle. The search was limited to peer-reviewed journal articles published in English. 

### 2.2. Eligibility Criteria

A PICOS approach was used to establish the inclusion and exclusion criteria [15], assessed by the review team.

***Population***: Patients requiring rehabilitation for a period of at least 1 week. All studies referring to rehabilitation were included in this review. We did not focus on any particular condition, since one of the aims of this study was to determine which type of patient/pathology could benefit most from telerehabilitation.

***Intervention***: Studies using communication technologies (smartphones, tablets, personal computers), apps and web-based platforms to deliver telerehabilitation services were included. Studies relying on the use of specific sensors (e.g., accelerometers), as well as non-specific games, virtual reality or active video games (e.g., Nintendo Wii, Microsoft Xbox Kinect) were instead excluded. 

***Control***: Traditional in-person rehabilitation or absence of care.

***Outcome measures***: Any type of outcome measure related to the International Classification of Functioning, Disability and Health (ICF). 

***Study design***: Randomized controlled trials (RCTs), explorative studies, observational studies; the sample had to include at least 10 patients.

### 2.3. Study Selection

All citations identified through the search strategy described above were uploaded on Rayyan in order to remove duplicates. A two-step screening process was carried out by three reviewers (EB, FV and OV), independently screening the first titles and abstracts for relevance against the eligibility criteria, and then proceeding with screening the full text of the selected articles. Disagreements were resolved between the reviewers, with the involvement of a fourth reviewer when necessary. The reference list of the studies thus included was also screened to identify additional papers appropriate for inclusion, according to the snowball principle.

The results from this search and the selection process are presented in a PRISMA flow diagram (Figure 1).

### 2.4. Quality Assessment

Due to the variety of publications included, we used the PEDro Scale to evaluate the methodological quality and risk of bias of RCTs [16], the STROBE Checklist for observational studies [17,18] and the CONSORT Checklist for feasibility and pilot studies [19,20].

### 2.5. Data Extraction

Data were extracted from the selected studies for inclusion and tabulated, including the following relevant information: participants’ characteristics (condition, age), intervention (type, number of sessions, length, frequency, duration), type of control, main outcomes and conclusions (including feasibility and effectiveness) (see Table 1, Table 2 and Table 3).

### 2.6. Ethical Approval and Reporting

For the present study, no ethics committee approval was necessary. 

This review was reported following the Preferred Reporting Items for Systematic Reviews and Meta-Analyses (PRISMA) recommendations [21].

## 3. Results

Twenty studies were included in the final analysis (see flow diagram in Figure 1). Of the included studies, 6 were RCTs, 10 were observational studies and 4 were exploratory studies (feasibility and pilot trials), including a total sample of 224,806 subjects: 209,218 with orthopedic complaints and 15,588 with non-orthopedic conditions. Of the latter, 3843 were patients with vestibular dysfunction, 1184 with neurological conditions, 369 with cardiovascular diseases, 362 with COVID-19 and 9830 with a variety of other pathologies (fibromyalgia, overweight and obesity, spinal disorders, pulmonary, oncology, oedema and other not-better-specified pathologies) (Figure 2). It is important to note that the number of orthopedic, vestibular and some of the “other pathologies” patients mainly come from one study alone [22].

**Table 1 ijerph-19-10325-t001:** Characteristics of the RCTs included in the review.

Author, Year, Country	Method. Quality	Population (N, Pathology, Age—Years)	Intervention (Type, Sessions, Length, Frequency, Total Duration)	Control	Delivery Method	Main Outcomes	Conclusion
Batalik et al. (2021) Czech Republic [23]	PEDro Scale 6/10	44 patients with coronary heart disease (CAD) (56.6 ± 7.3)	n = 21 Home-based cardiac telerehabilitation w/wrist monitor and telemonitoring/coaching; exercise 3×/week for 60 min at 70–80% of heart rate reserve for 12 weeks	n = 23 Traditional center-based cardiac rehabilitation (same intervention but supervised in person)	Mixed	Cardio- respiratory fitness (CRF) and health-related quality of life (HRQL)	At 1 year follow-up, home-based cardiac telerehabilitation was more effective than center-based cardiac rehabilitation in maintaining long-term CRF levels (*p* = 0.047). No statistically significant difference between the two groups for HRQL.
Gonzalez- Gerez et al. (2021) Spain [24]	PEDro Scale 8/10	38 COVID-19 patients with mild to moderate symptoms in the acute stage (28–53)	n = 19 7-day pulmonary rehabilitation (10 breathing exercises, performed daily at home); additional two videoconferences with a physiotherapist and daily text messages	n = 19 No intervention	Mixed	6 min walk test (6MWT), dyspnea (MD12), 30 s sit-to-stand test (30STST), Borg Scale	Significant differences were found for all outcomes in favor of the intervention group, with 90% adherence. A 1-week telerehabilitation program based on respiratory exercises is effective, safe and feasible in COVID-19 patients with acute mild–moderate symptoms.
Hernando- Garijo et al. (2021) Spain [25]	PEDro Scale 7/10	34 women with fibromyalgia (53.44 ± 8.8)	n = 17 50 min sessions over 15 weeks (two sessions/week) of telerehabilitation (low-impact rhythmic movements, guided by video)	n = 17 No additional intervention (asked to maintain the same medical prescription during the study)	Mixed	Pain intensity, mechanical pain sensitivity, psychological distress	A telerehabilitation aerobic exercise program yielded statistically significant improvements in pain intensity (*p* = 0.022), mechanical pain sensitivity (*p* < 0.05) and psychological distress (*p* = 0.005) compared to the control group, which showed no statistically significant changes in any variable (*p* > 0.05).
Li et al. (2021) China [26]	PEDro Scale 8/10	120 formerly hospitalized COVID-19 survivors with remaining dyspnea complaint (18–75)	n = 59 TERECO: unsupervised home-based 6-week exercise program; three–four sessions/week of breathing, aerobic exercise and strengthening exercises; delivered via smartphone app and monitored via heart rate telemetry	n = 61 No intervention (only received short educational instructions at baseline)	Mixed	6MWT, lower limb strength, pulmonary function (spirometry), HRQL, dyspnea. Outcomes assessed at week 6 (post-treatment) and 28 (follow-up)	Results demonstrated the superiority of TERECO over no rehabilitation for 6MWD (*p* < 0.001), lower limb strength (*p* < 0.001) and HRQOL (*p* = 0.004).
Ozturk et al. (2022) Turkey [27]	PEDro Scale 5/10	41 patients with BMI > 25 (18–65)	n = 21 Exercise training with remote live connection (warm-up, trunk stabilization and breathing exercises) supervised by a physiotherapist (3×/week for 6 weeks)	n = 20 Only informed about the importance of exercise for one session	Synchronous	Physical fitness (Senior Fitness Test protocol), HRQL (SF-36), evaluated at baseline and after 6 weeks (post-treatment)	All parameters were statistically significantly different in favor of the telerehabilitation group (*p* < 0.05). Exercise training via telerehabilitation during the COVID-19 pandemic was effective, safe and feasible in overweight and obese individuals (BMI > 25).
Rodríguez- Blanco et al. (2021) Spain [28]	PEDro Scale 8/10	77 subjects with COVID-19 in the acute stage (39.40 ± 11.71)	n = 29 Breathing exercises n = 26 Strengthening exercises Both groups performed exercises 1×/day for 14 days; they were taught on day 1 via videoconference; reinforced 1×/week; additional daily text message	n = 22 No intervention	Mixed	Visual analog scale for fatigue, 6MWT, 30STST, dyspnea (MD-12), Borg scale Assessed at baseline and 14 days later	The strength and breathing groups achieved significant improvements in fatigue, dyspnea, perceived effort and physical state compared to control group (*p* < 0.05). The greatest benefits were found for dyspnea and aerobic capacity in the breathing group (*p* < 0.001).

**Table 2 ijerph-19-10325-t002:** Characteristics of the observational studies included in the review.

Author, Year, Country	Method. Quality	Population (Pathology, Age—Years)	Intervention (Type, Sessions, Length, Frequency, Total Duration)	Comparison	Delivery Method	Main Outcomes	Conclusion
Cancino- López et al. (2021) Chile [29]	STROBE Checklist 18/22	50 COVID-19 patients (54.1 ± 15.4)	24 exercise sessions of 50–60 min each (10 min warm up, 25 min resistance training, 10 min aerobic training, 5 min cool down), 2–3×/week, via video calls	(No comparison)	Synchronous	Functionality (Barthel’s index) and physical fitness (2 min step test), elbow flexion—one repetition maximum (1RM), short physical performance battery, hand grip strength, 30 s chair stand, skeletal muscle index, body fat percentage, resting pulse, arterial blood pressure and pulse oximetry	24 sessions of in-home telerehabilitation exercise program promoted the recovery of physical independence, with significant improvements in functionality and physical fitness (*p* < 0.0001).
De Marchi et al. (2020) Italy [30]	STROBE Checklist 19/22	19 patients with ALS (51.48)	Televisit of 80–120 min, 3×/month for 3 months (multidisciplinary approach: neurologist, dietician, psychologist, physiotherapist)	(No comparison)	Synchronous	Anxiety and depression (HADS and ALSAQ-40), functional status (ALSFRS-R, Barthel scale), exertion (Borg scale) and pain intensity (VAS)	ALS patients managed by telemedicine received a comparable quality of care to those seen via traditional face-to-face methods; this needs to become an integrated platform for delivering high-quality tertiary ALS care.
Lamberti et al. (2021) Italy [31]	STROBE Checklist 21/22	66 patients with peripheral artery disease (PAD) (72)	2 × 8 min daily sessions of slow intermittent in-home walking. Additional regular phone calls to check in on patients	(No comparison)	Synchronous	6MWD, pain-free walking distance, body weight blood pressure, ankle–brachial index	Pain-free walking distance improved significantly (*p* < 0.001), body weight decreased, while 6MWD, blood pressure and ankle–brachial index remained stable. A structured in-home walking program guided by phone was adhered to by patients with PAD and improved their mobility.
Milani et al. (2021) Italy [32]	STROBE Checklist 19/22	23 patients with physical disabilities (44–70.6)	Physiotherapist-led telerehabilitation program with customized exercises; 1 h sessions 2–3 times/week from March to May 2020, delivered in real time via Skype	No tele-rehabilitation	Synchronous	Feasibility and acceptability	Telerehabilitation was a feasible solution, with high adherence and well accepted by patients.
Negrini et al. (2020) Italy [33]	STROBE 16/22	1207 patients with spinal disorders, (3–18)	Teleconsultations and telephysiotherapy delivered over 3 weeks (15 working days)	Traditional in-person physiotherapy (13 working days)	Mixed	Number of services provided and patient satisfaction	Telephysiotherapy was feasible and allowed health professionals to continue providing outpatient services with a high patient satisfaction, reducing face-to-face contact and the need for travel to a minimum.
Oprandi et al. (2021) Italy [34]	STROBE Checklist 19/22	13 children and young adults with acquired brain injury (ABI) (10.7)	Neuropsychological and speech telerehabilitation sessions (2×/week for 10 weeks)	(No comparison)	Synchronous	Feasibility and acceptability	Feasibility and acceptability of synchronous telerehabilitation for young patients with ABI was demonstrated. Telerehabilitation can be a successful intervention for this population.
Patel et al. (2021) India [35]	STROBE Checklist 16/22	47 patients (23 cardio-vascular, 15 pulmonary, 9 oncology) (61.2 ± 12.5)	Exercise telerehabilitation program (5–10 min warm-up, 20–25 min aerobic and strengthening exercises; plus +30 min brisk walk); 3×/week for 1 month	(No comparison)	Synchronous	6MWT, HRQL (FACIT), daily step count	A short-term, supervised telerehabilitation program yielded significantly positive effects on 6MWT (*p* = 0.0418) and HRQL (*p* = 0.0313) in cardiac, pulmonary and oncology patients during COVID-19.
Romano et al. (2021) Italy [36]	STROBE Checklist 20/22	13 patients with Rett syndrome (RTT) (17 y 11 m)	3-month home-based, individualized rehabilitation program of motor activities, remotely supervised via Skype calls	(No comparison)	Synchronous	Gross motor function	A total of 76.9% of participants significantly increased their gross motor function. A high level of usefulness, adherence and general satisfaction was observed. Findings strongly support the implementation of telerehabilitation programs for this population.
Sakai et al. (2020) Japan [37]	STROBE Checklist 18/22	43 COVID-19 patients undergoing rehabilitation (21–95)	n = 18 Remote rehabilitation via videocalls on iPad, with exercises to develop strength, endurance, range of motion and flexibility. Daily 20 min sessions for 1 month	n = 25 In-person rehabilitation with exercises to develop strength, endurance, range of motion and flexibility	Synchronous	ADLs (Barthel Index), mobility scores	The remote rehabilitation group had significantly higher scores in the Barthel Index than the in-person group. Remote rehabilitation is an effective and safe modality and can facilitate rehabilitation in various situations, including patients that can be treated at a distance.
Werneke et al. (2021) USA [22]	STROBE Checklist 20/22	222,680 patients with a variety of conditions (55 ± 18)	Telerehabilitation (6% of all episodes of care)	Traditional in-person visits	Synchronous (60%), asynchronous (21%), mixed (19%)	Physical function, number of visits, patient satisfaction, telerehabilitation frequency and modes	Telerehabilitation rate was 6%, decreasing from 10% to 5% between the second and third quarters of 2020. The rate of patients very satisfied with their treatment was 3% higher for no telerehabilitation. More studies are needed to understand what facilitates and inhibits the use of telerehabilitation by rehabilitation therapists in order to promote it when appropriate.

**Table 3 ijerph-19-10325-t003:** Characteristics of the exploratory studies (feasibility and pilot studies) included in the review.

Author, Year, Country	Method. Quality	Population (Mean Age, Pathology)	Intervention (Type, Sessions, Length, Frequency, Total Duration)	Comparison	Delivery Methods	Main Outcomes	Conclusion
Lowe et al. (2021) UK [38]	CONSORT Checklist 17/25	21 patients with MS (18+)	LEAP-MS Online Intervention (3 months) delivered via Zoom calls and a web-based online physical activity tool	(No comparison)	Mixed	Fatigue (MFIS) impact of MS(MSIS-29), HRQOL (EQ-5D-5 L), impact of ill health on participation and activities (OxPAQ), self-efficacy (UW-SES-SF), impression of change (PGIC)	This feasibility study allowed meeting the needs of people with MS during the COVID-19 pandemic.
Martin et al. (2021) Belgium [39]	CONSORT Checklist 17/25	27 patients with COVID-19 (61.5 ± 10.5)	n = 14 telerehabilitation via videoconference; 50 min sessions 3×/week for 6 weeks (30 min endurance exercises, 20 min strengthening exercises); Borg Scale: 6	n = 13 No intervention (patients who refused the telerehabilitation intervention)	Synchronous	Functional exercise capacity (1 min STST), SpO2 at rest	At 3 months follow-up, improvements were significantly and clinically greater in the telerehabilitation group (*p* = 0.005). The feasibility and effectiveness of a simple telerehabilitation program were verified.
Nakayama et al. (2020) Japan [40]	CONSORT Checklist 16/25	236 patients hospitalized for heart failure (HF) (59)	n = 30 remote cardiac rehabilitation (CR)	n = 69 outpatient CR n = 137 non-CR	Mixed	HRQL (EQ5D) 30 days after discharge; Number of emergency readmissions (%)	Emergency readmission rate within 30 days of discharge was lower in the remote CR group than in the non-CR group (n = 137) (*p* = 0.02). HRQL score was higher in the remote CR group than in the outpatient CR group (*p* = 0.03) 30 days after discharge. The remote CR program can be a good alternative to outpatient CR.
Tanguay et al. (2021) Canada [41]	CONSORT Checklist 15/25	Seven COVID-19 patients (49–80)	Physiotherapist- led telerehabilitation intervention delivering a pulmonary telerehabilitation program (2×/week for 8 weeks)	(No comparison)	Mixed	Severity of pulmonary symptoms (CAT), HRQL (EQ-5D-5L, EQ-VAS)	All participants increased their quality-of-life scores by at least 10 points. Eight weeks of telerehabilitation seem to improve symptoms, quality of life and return to physical activities in COVID-19 patients.

The characteristics of patients and the different interventions are presented in Table 1 for RCTs, Table 2 for observational studies and Table 3 for exploratory studies. 

The most targeted population were patients with orthopedic complaints (93.1%, see Figure 2), while the age of subjects included in the total sample ranged from 3 to 95 years. 

The most utilized telerehabilitation strategy involved synchronous (i.e., simultaneous exchange of information via video or audio calls) (53%) or mixed modes (i.e., using both synchronous and asynchronous modes) (43%), mainly using video and audio calls, as well as text messages and links to educational content. Note that the asynchronous mode (i.e., remote visits, not in real time, via text messaging, applications or links to educational materials) was barely used (1% of the treated patients; only used in one arm of one study [22]).

Telerehabilitation interventions were typically delivered via commonly available technologies, such as smartphones or personal computers, and free videoconferencing platforms, including Google Meet, Zoom, Skype and Whatsapp (Figure 3).

For the sake of clarity, to discuss the different clinical applications, we regrouped the studies according to the different targeted pathologies/conditions: COVID-19, cardiovascular, neurological and other conditions (see Figure 3 for a schematic breakdown of the total number of subjects, divided by condition and study, as well as the technologies used).

Concerning the quality of the included studies, the RCTs ranged from “fair” to “good” (mean score: PEDro Scale 7/10; range: 5/10–8/10; see Appendix A); the quality of the observational studies was moderate to very good, with a mean score of 17/22 on the STROBE checklist (range: 16/22–21/22; see Appendix A); lastly, the exploratory studies showed moderate quality (mean score: CONSORT checklist 16/25; range: 15/25–17/25; see Appendix A).

### 3.1. COVID-19

Seven studies analyzed the use of telerehabilitation with patients affected by COVID-19. Of these, three were RCTS, two were observational studies, and the last two were exploratory studies; although interventions varied slightly from one study to the other, telerehabilitation proved to be an effective, safe and feasible modality to facilitate the recovery of these patients.

More specifically, the RCT by Gonzalez-Gerez et al. (2021) [24] found that delivering breathing exercises via telerehabilitation was a promising, safe and effective strategy for improving physical performance, dyspnea and perceived effort in patients with COVID-19 in the acute stage. The breathing exercises were carried out by patients once a day for 1 week at home, and the program was reinforced by a physiotherapist via videoconference; patients also received a daily text message to increase adherence. The intervention group achieved statistically significant changes and showed much higher effect sizes compared to the controls in all outcomes, supporting the clinical relevance of this telerehabilitation intervention.

Similar results were found in another RCT [26] where authors investigated the effects of a 6-week unsupervised home-based exercise program. It consisted of breathing, aerobic and lower limb muscle strength exercises, delivered to COVID-19 survivors via smartphone and remotely monitored with heart rate telemetry. Outcomes assessed at week 6 (post-treatment) and 28 (follow-up) showed the superiority of the intervention regarding exercise capacity, lower limb muscle strength and quality of life. Adherence was satisfactory, with no serious adverse events observed.

Moreover, in 2021, Rodríguez-Blanco et al. [28] published the results of another RCT comparing the effectiveness of two different exercise-based programs (strengthening and breathing exercises) delivered through telerehabilitation in subjects with COVID-19. At the end of the 14-day intervention, statistically significant differences were found between the two intervention groups and controls in all variables (fatigue, dyspnea, perceived effort and physical condition), although the greatest benefits for dyspnea and aerobic capacity were found in the breathing exercises group.

Regarding the observational studies, the one by Cancino-López et al. (2021) [29] examined COVID-19 patients who completed a 24-session telerehabilitation program consisting of 50–60 min of resistance and aerobic training, performed 2–3 times a week. At the end of the program a significant increase in function, physical fitness, aerobic capacity and upper and lower body strength was found, indicating that a home telerehabilitation program promotes recovery in people with COVID-19. 

Another observational study [37] retrospectively described the effectiveness and risk management of remote rehabilitation for COVID-19 patients. At discharge, patients in the remote rehabilitation group had significantly higher scores compared to the in-person rehabilitation one, but it must be noted that patients in the latter group also had more severe symptoms. No adverse events were observed, and remote rehabilitation turned out to be an effective and safe modality that could facilitate rehabilitation of patients with COVID-19.

The effects of a telerehabilitation program in COVID-19 patients was further investigated by Martin et al. (2021) [39]. A pulmonary rehabilitation program was delivered via videoconferencing by an experienced physiotherapist; patients performed home-based exercises 2 times a week for 6 weeks, each session involving 30 min of endurance exercises and 20 min of upper and lower body muscular strengthening. No adverse events were reported. The results showed statistically and clinically significant improvements in functional exercise capacity, dyspnea, oxygen saturation and heart rate, verifying the feasibility and effectiveness of a telerehabilitation program for the recovery of COVID-19 patients.

Lastly, the feasibility of telerehabilitation in patients with COVID-19 was also assessed by a pre-experimental, pre–post pilot study by Tanguay et al. (2021) [41]. A 2-stage 8-week-long telerehabilitation intervention was administered by a respiratory physiotherapist; the first stage included a bi-weekly remotely supervised session of breathing, aerobic and strengthening exercises and intensive patient education on self-management skills; the second one instead involved only a once-weekly remotely supervised session for consolidation. Patients were also advised to carry out a daily 30 min unsupervised cardiorespiratory and breathing exercise session and a weekly 60 min unsupervised strengthening session (twice weekly in the second phase). At the end of the intervention, patients showed a clinically significant improvement in functional health and quality of life, suggesting that 8 weeks of telerehabilitation improved recovery in COVID-19 patients.

### 3.2. Neurological Diseases

Four observational and one feasibility study aimed to assess the effects of implementing telerehabilitation with patients affected by neurological conditions. The results showed that telerehabilitation was a feasible and well-accepted solution that effectively improved the desired outcomes.

One of the observational studies [30] demonstrated that the management of amyotrophic lateral sclerosis (ALS) with telemedicine could be as effective as face-to-face healthcare. Patients were evaluated and treated via televisits by a multidisciplinary team, also including a physiotherapist. A significant stabilization of patients’ quality of life related to muscle strength, motor disability and respiratory failure was observed, and all patients were satisfied with this approach.

Another study [38] described the effects of telerehabilitation in patients with multiple sclerosis (MS). The authors aimed to assess whether the lifestyle, exercise and activity package for people living with progressive multiple sclerosis (LEAP-MS) was feasible during quarantine. Face-to-face coaching sessions were substituted with video and telephone consultations. The findings showed that a revised LEAP-MS delivered via telerehabilitation was indeed a feasible solution, with a shorter waiting time for participants.

The study by Romano et al. (2021) [36] instead presented the effects of a remotely supervised rehabilitation program for people with Rett syndrome (RTT) on gross motor function. Each patient received a personalized program, including 1 daily non-consecutive hour of activity, 5 days a week for 3 months. Two 1 h videoconference calls were organized at month 1 and 2 to check in on patients. This remotely supervised rehabilitation program increased adherence and effectively improved motor function.

On the other hand, Milani et al. (2021) [32] aimed to assess the feasibility and acceptability of a telerehabilitation program during the COVID-19 lockdown period in Italy, between March and May 2020, in a group of adults with neurological and orthopedic disabilities. Patients participated in two–three sessions/week, each one lasting roughly 1 h, performing exercises as instructed by a specialized physiotherapist via synchronous Skype videocalls, using either mobile phones, tablets or computers. Analyses showed high adherence with no dropouts and, even though participants reported preference for in-person rehabilitation, they appreciated the flexibility provided by telerehabilitation, also proving the feasibility of a telerehabilitation program during the COVID-19 emergency.

Similarly, the last study [34] assessed the feasibility and acceptability of telerehabilitation during the COVID-19 pandemic in a group of children and young adults with acquired brain injury. Participants performed two sessions a week of speech and neuropsychological telerehabilitation via video calls for 10 weeks. No technical or clinical obstacles were observed, and the synchronous telerehabilitation intervention was found to be a feasible and well-accepted intervention, suggesting that it can be a successful solution for young people with ABI. 

### 3.3. Cardiovascular Diseases

The studies included in the present review focusing on cardiovascular patients were three in total: one RCT, one observational and one exploratory study. The findings demonstrated that remotely delivered cardiovascular rehabilitation is not only feasible but also effective at improving functional health-related outcomes in this group of patients.

Batalik et al. [23] assessed the long-term effects of a home-based cardiac telerehabilitation (HBCT) program on patients with coronary artery disease. This program consisted of 60 min exercise sessions three times a week for 12 weeks, at an intensity of 70–80% of heart rate reserve, and a once-weekly telephone consultation. Data were collected post-intervention and at 1-year follow-up. Significant improvements in quality of life and cardiorespiratory fitness were reported, comparable to the same program delivered in person, demonstrating that cardiovascular telerehabilitation is a feasible and effective solution to improve health-related outcomes in these patients.

The observational study by Lamberti et al. (2021) [31], on the other hand, focused on evaluating the effects of an in-home walking rehabilitation program in patients with peripheral artery disease (PAD). This program consisted of two phases: a center-based phase of circa-monthly visits to the hospital and a home-based one involving twice-daily 8 min sessions of slow intermittent in-home walking. Patients also received regular phone calls from the team. The results showed improvements in pain-free walking distance and a reduction in body weight, indicating that a home-based walking program guided by phone was adhered to by participants and promoted mobility in patients with PAD.

The last study [40] instead prospectively investigated remote cardiac rehabilitation (CR) in patients diagnosed with heart failure. The intervention group received DVD guides for home-based CR, as well as telephone consultations every 2 weeks for 5 months. The results showed that remote CR was as effective as outpatient CR for improving post-discharge short-term prognosis of subjects with heart failure, suggesting that remote CR can be a suitable alternative to traditional outpatient CR.

### 3.4. Other Conditions

Among the studies included in the present review, one RCT studied the effect of telerehabilitation on patients with fibromyalgia and another on overweight and obese patients. In addition, an observational study focused on patients with spinal disorders, and two more included patients with a variety of conditions (cardiovascular, pulmonary, oncology, orthopedic, stroke, edema and vertigo).

The RCT by Hernando-Garijo et al. (2021) [25] analyzed the immediate effects of a telerehabilitation exercise program in women with fibromyalgia. The intervention consisted of twice-weekly 50 min stretching and aerobic exercise sessions for 15 weeks and a once-weekly video call. Participants achieved statistically significant improvements in pain intensity, mechanical pain sensitivity and psychological distress compared to no intervention, proving the effectiveness of such telerehabilitation intervention.

Exercise training delivered through telerehabilitation was also an effective, safe and feasible approach in overweight and obese individuals in the study by Ozturk et al. (2022) [27]. A remotely delivered program, including warm-up, trunk stabilization and breathing exercises, 3 days a week for 6 weeks, indeed yielded statistically significant improvements in physical fitness and quality of life.

In the observational study by Negrini et al. (2020) [33], which investigated the feasibility and acceptability of telemedicine as a substitute for outpatient services during the COVID-19 pandemic in Italy, teleconsultations and telephysiotherapy were provided to 1207 patients with spinal disorders over a period of 15 working days. The results suggested that this approach was feasible and allowed healthcare professionals to provide outpatient services with a very high patient satisfaction, making it a viable alternative to reduce face-to-face contact and the need for travel to a minimum.

Moreover, the short-term effects of telerehabilitation on cardiac, pulmonary and oncology patients were evaluated in a study by Patel et al. (2021) [35]. A telerehabilitation involving home-based exercise interventions 3 days a week was delivered via videoconference. Each 30 min session included a warm-up, followed by aerobic and strengthening exercises; patients were also invited to walk at least 30 min per day. According to the results, this short-term, supervised telerehabilitation program had significantly positive effects on walking performance and quality of life.

One last study [22] retrospectively described baseline patient characteristics for episodes of care offered during COVID-19 using telerehabilitation, as well as modes of delivery and frequency by condition. The results showed that patients using telerehabilitation were more likely to be younger and live in large metropolitan areas. Telerehabilitation was administered equally across different orthopedic body parts, with a lower use for non-orthopedic conditions (e.g., stroke, edema, vestibular dysfunction). The preferred mode was synchronous video or audio call.

## 4. Discussion

This systematic review aimed to summarize and analyze the telehealth solutions that were proposed to provide rehabilitation services to a variety of patients during the COVID-19 pandemic.

Telerehabilitation seems to be a feasible and effective option to remotely treat and support patients with a variety of conditions during the COVID-19 pandemic. The most used strategies to deliver telerehabilitation interventions were synchronous (via video and audio calls) and mixed modes (with the addition of text messages and educational links), leveraging commonly available technological devices, such as smartphones and personal computers, and free videoconferencing platforms, such as Google Meet, Zoom, Skype or Whatsapp. Such convenient and relatively inexpensive tools indeed have the ability to make telerehabilitation considerably accessible and feasible for both patients and clinicians. 

The seven studies that investigated the implementation of telerehabilitation with COVID-19 patients [24,26,28,29,37,39,41] found that patients had significantly improved their functional and health-related outcomes. The results were satisfactory and comparable to in-person rehabilitation, with the advantages of reducing the need to travel and the risk of infection. It is to be noted that even though the number of studies on COVID-19 patients represents about 37% of all the studies included, the number of patients with COVID-19 in this review is relatively limited (n = 362 or 2–3% of the total number of patients). This can be explained by the fact that COVID-19 is a relatively new condition, and the use of telerehabilitation with these patients was therefore a novel and understandably scarcely diffused solution (unlike, for example, orthopedic, cardiovascular or neurological conditions, where telerehabilitation was already in use prior to the current pandemic [42]). It is important to also highlight that rehabilitation protocols generalizable to all COVID-19 patients are difficult to establish, since these patients may present with very different degrees of symptoms. In fact, the rehabilitation plans for COVID-19 patients need to be customized according to each patient’s unique characteristics (e.g., age, sex, comorbidities, lifestyle, occupation and hobbies) [43,44]. Special attention, however, should always be paid to restoring the respiratory, cardiovascular and motor functions for an optimal recovery of these patients [43,44]. Interestingly enough, a relatively recent systematic review recommends, whenever possible, to preferably use telerehabilitation in outpatient rehabilitation settings [44].

The effectiveness and feasibility of telerehabilitation was also seen in patients with various neurological disorders, with results showing significant improvements in patients affected by amyotrophic lateral sclerosis, multiple sclerosis, Rett syndrome, acquired brain injury and other neurological disabilities [30,32,34,36,38]. Remote rehabilitation indeed made it possible to reach patients unable to travel and to overcome the need for recurrent outpatient visits.

The use telehealth to manage cardiovascular diseases was already known before the COVID-19 surge, but its adoption has significantly increased throughout the pandemic. The studies included in this review regarding patients with cardiovascular diseases also found that remote rehabilitation was a feasible, effective, as well accepted modality in patients with heart failure, coronary artery disease and peripheral artery disease [23,31,40].

Lastly, telerehabilitation also appeared to be a feasible, effective and generally well-accepted intervention in patients with a variety of other conditions, including orthopedic complaints, vestibular dysfunction, fibromyalgia, spinal disorders, stroke, oedema, as well as pulmonary, oncological, overweight and obese patients [22,25,27,33,35].

### 4.1. Limitations and Strengths

Among the limitations of this review, one is the lack of uniformity in the terminology used to characterize the many telehealth solutions currently being evaluated in research. As a result of the relative scarcity of published research on this topic, we included studies examining a variety of application and intervention modalities or durations. Additionally, it cannot be excluded that some publications might have been involuntarily missed during the selection process or that some potentially relevant studies might have been excluded due to the application of the chosen inclusion and exclusion criteria. Furthermore, considering that the mean quality of the studies included was moderate but not high and that their designs were considerably heterogeneous, strong recommendations cannot be provided based on the findings, and such limitations must be accounted for when drawing conclusions. 

The research methodology used in the present review, however, was rigorous, and the analysis of the publications included was thorough. In addition, the work hereby performed allowed answering the research questions, providing useful insights on the effectiveness and feasibility of the use of telehealth solutions to deliver remote rehabilitation, allowing the continuity of care for a variety of patients during the COVID-19 pandemic. Lastly, another important strength is the external validity of this review. Considering that the subjects included varied considerably in terms of age and health conditions, the findings can thus be more easily generalized.

### 4.2. Implications for Clinical Practice and Research

Telerehabilitation appears to be a feasible and effective strategy to ensure the remote delivery and continuity of rehabilitation services to a variety of patients during the COVID-19 pandemic. Where appropriate, its implementation should therefore be considered in clinical practice. This could indeed guarantee not only a reduction in the risk of infection but also in the need to travel, especially in case of regular outpatient visits, while guaranteeing the quality of care.

The delivery of telerehabilitation services can be carried out easily via video and audio calls and eventually supplemented by text messages or links to educational content, using commonly available devices, such as smartphones, tablets and laptops, and free videoconferencing tools, such as Google Meet, Zoom, Skype or Whatsapp.

However, the moderate quality of the studies included and their heterogeneity with respect to both the targeted populations and the selected modes of delivery highlight the need for further large, high-quality RCTs to strengthen these findings; more robust evidence is indeed needed regarding the use of telerehabilitation for the remote continuance of healthcare services to patients with very diverse health conditions.

Additionally, more work is also required to evaluate not only the effectiveness but also the cost effectiveness of telerehabilitation compared to in-person care, evaluating the economic implications of offering rehabilitation services via ICT. In fact, physiotherapy is typically used extensively, which also makes it extremely expensive, with aggregate costs reaching USD hundreds of millions annually, and rising [45]. However, if telerehabilitation could yield results comparable to traditional face-to-face rehabilitation but in a more cost-effective manner, this strategy could be further promoted as a viable alternative to deliver rehabilitation services even after the COVID-19 pandemic subsides. 

Several concerns, however, must be solved before these solutions may be employed in everyday practice. The first and most critical is the recognition of eHealth and telehealth applications as medical equipment. In June 2020, the United States Food and Drug Administration (FDA) approved the commercialization of the first game-based digital therapy device for children with attention deficit hyperactivity disorder (ADHD). The mHealth, EndeavorRx, is suggested to enhance the attention function, as evaluated by computerized testing. This is the first digital treatment intended to alleviate symptoms associated with ADHD, as well as the first game-based therapeutic to obtain FDA marketing authorization for any ailment, making it an important milestone for the development, adoption and acknowledgment of telehealth in clinics [46].

It is important to stress, however, that the majority of steps implemented during the pandemic may be transitory, and further efforts are therefore needed in this area to facilitate the use of telerehabilitation even after the current crisis subsides. For example, it will be necessary to alter the nomenclature of treatments, since mobile solutions are now classified alongside pharmaceuticals, posing challenges for their validation and payment [47].

Additionally, other significant barriers to telehealth implementation that need to be taken into consideration are budgetary constraints, data privacy and reimbursement issues and a lack of understanding and familiarity with the use of (new) technology [48,49]. With particular regard to the last point, efforts must also be made to educate healthcare professionals, since they must be thoroughly knowledgeable about the available communication technologies to then instruct and encourage patients to use them for telehealth and telerehabilitation services.

## 5. Conclusions

The present review offered valuable insights regarding the use of telehealth to remotely deliver rehabilitation services to a variety of patients during the COVID-19 pandemic, thus guaranteeing the continuity of care.

Considering that the actual extent of the use of telehealth for rehabilitation purposes was still unclear, the present review aimed to summarize the solutions that have been proposed to offer remote rehabilitation to patients during the COVID-19 pandemic. The current studies of moderate quality showed telerehabilitation to be a feasible and effective option to allow the continuity of care for a variety of patients and conditions; indeed, it would appear that telerehabilitation, mainly delivered through video and audio calls, has allowed maintaining the quality of rehabilitation while reducing the risk of infection and the burden of travel.

Where appropriate, the implementation of telerehabilitation in clinical practice could therefore be considered an alternative or complementary option to traditional in-person care.

Future research is, however, needed to confirm these findings, providing stronger evidence regarding the most appropriate use of telerehabilitation in clinical practice, as well as its cost effectiveness, in order to promote and leverage its implementation not only during the current pandemic but also beyond.

## Figures and Tables

**Figure 1 ijerph-19-10325-f001:**
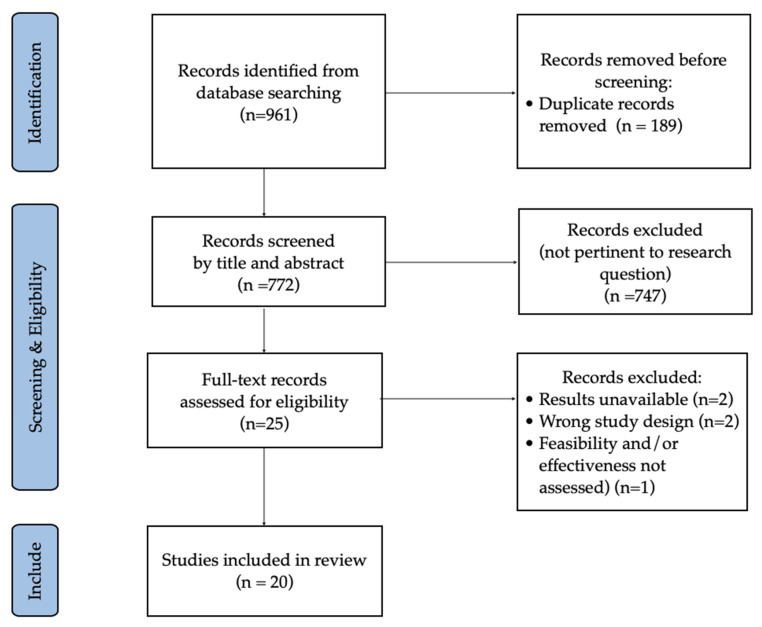
PRISMA flow diagram of study selection.

**Figure 2 ijerph-19-10325-f002:**
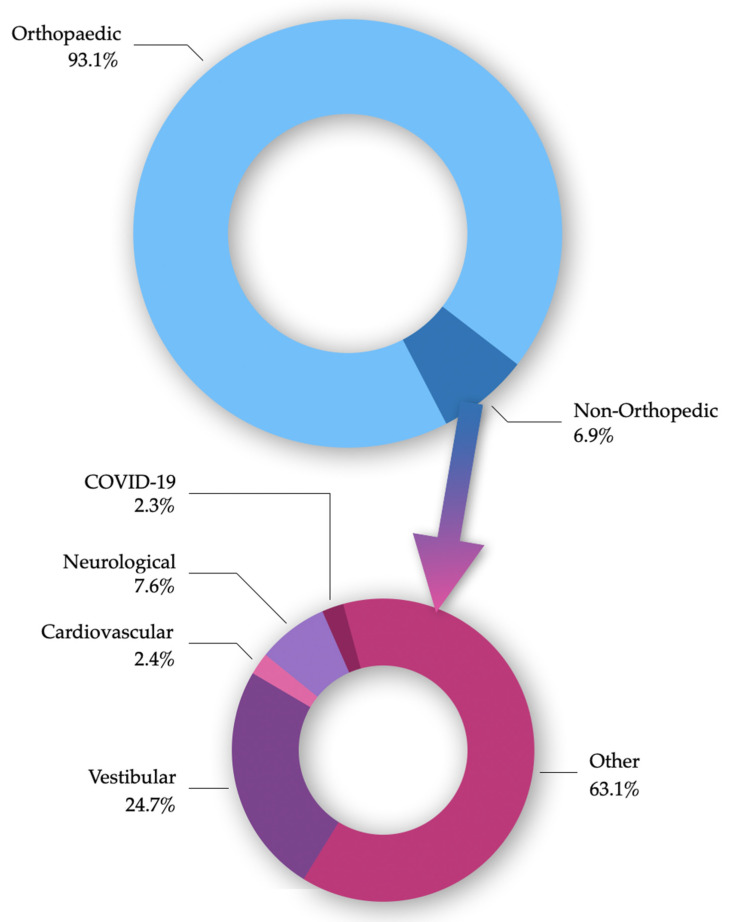
Percentages of the total number of subjects included in the present review with either orthopedic or non-orthopedic complaints. Non-orthopedic complaints were further divided by category (“Other” refers to a group of various pathologies, including: oedema, fibromyalgia, overweight and obesity, spinal disorders, acquired brain injury, pulmonary, oncology and not-better-specified diseases).

**Figure 3 ijerph-19-10325-f003:**
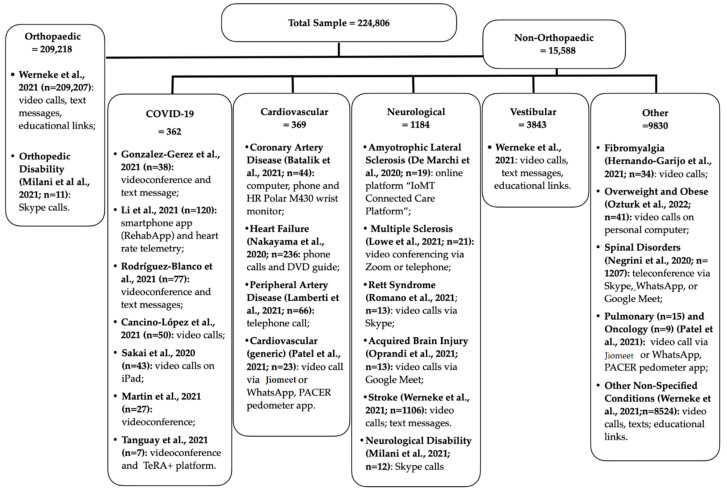
Schematic illustration of the total sample of the participants and the technologies used, divided by condition and study [22,23,24,25,26,27,28,29,30,31,32,33,34,35,36,37,38,39,40,41].

## Data Availability

Not applicable.

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
