# Peer review of "Using Telehealth to Guarantee the Continuity of Rehabilitation during the COVID-19 Pandemic: A Systematic Review"

_ijerph, 2022, doi:10.3390/ijerph191610325_

Round 1

Reviewer 1 Report

Overall, I really enjoyed reading this article. The article has a lot of great information. It provides a good review of telerehabilitation. Here are a few minor suggestions: 

1.       Introduction – there was some utilization of tele-rehab prior to the pandemic as well. Which exponentially increased during the pandemic. If the authors can provide some details on the prior vs current usage that would add value to the introduction section.

2.       I do not see a complete figure 4, it is also blurry. Please provide a proper good quality image.

3.       This review was not specific to certain diagnoses or certain techniques. This review cannot generalize the quality of rehabilitation provided based on these articles. Certain types of patients or diagnoses might be more suitable for tele rehab compared to others. This article included a lot of different study designs and analyze them under one broad category – different study designs have their own biases which introduce bias in the analysis and overall outcome and conclusion.  All such aspects should be mentioned in the limitations.

4.       In the conclusion the authors called this a moderate -quality evidence. I do not think the authors performed any qualitative analysis to provide such a comment.

5.       The same way the abstract, says “Telerehabilitation seems a feasible and effective option to remotely deliver high-quality rehabilitation using …” – I do not think it is appropriate to comment on the quality of the rehabilitation. 

Author Response

Hasselt, 5th August 2022

I wish to thank the Guest Editor and the Reviewers for taking the time to read our manuscript and provide us with valuable suggestions. All the comments received have been duly addressed, adjusting some parts of the manuscript in order to strengthen it, based on the Reviewers’ remarks. I appreciate the opportunity to make such revisions, trusting these will be satisfactory to both Reviewers.

The answers to the comments received can be found in the attached file, written in red. The changes made in the manuscript, instead, have been highlighted in yellow. 

Sincerely,

Elisabetta Brigo

elisabetta.brigo@uhasselt.be

Reviewer 2 Report

It was a pleasure to review this article.

My congratulations to the authors for their work, however, I have some considerations to make:

In the line 82: “The search was limited to peer-reviewed jornal”. How as controlled this aspect?

Regarding the Eligibility criteria: what was the criterion for considering programs with at least 1 week? (line 85).

Double blinding in review studies is essential. In systematic review studies, blinding in the selection of articles and in the extraction of information is essential. Resorting another investigator in case of disagreement. These aspects are mandatory and must be described in the methodology.

In the Tables, are suggested that the articles be cited with the proper numerical reference.

Figure 4 cannot be fully visualized.

The article summarizes the results well, however it is necessary to identify the evidence obtained in each of the studies. This was the purpose of the present work, mentioned in the introduction: to assess the effectiveness and validity of interventions.  It would be interesting to see this aspects more clearly written in the article.

Author Response

(The authors gave the same response as above.)
